# PM_2.5_ Pollution Strongly Predicted COVID-19 Incidence in Four High-Polluted Urbanized Italian Cities during the Pre-Lockdown and Lockdown Periods

**DOI:** 10.3390/ijerph18105088

**Published:** 2021-05-11

**Authors:** Ourania S. Kotsiou, Vaios S. Kotsios, Ioannis Lampropoulos, Thomas Zidros, Sotirios G. Zarogiannis, Konstantinos I. Gourgoulianis

**Affiliations:** 1Faculty of Nursing, University of Thessaly, GAIOPOLIS, 41110 Larissa, Thessaly, Greece; 2Respiratory Medicine Department, Faculty of Medicine, University of Thessaly, BIOPOLIS, 41110 Larissa, Thessaly, Greece; johnis08@yahoo.gr (I.L.); kgourg@med.uth.gr (K.I.G.); 3Department of Physiology, Faculty of Medicine, University of Thessaly, BIOPOLIS, 41500 Larissa, Thessaly, Greece; szarog@med.uth.gr; 4Metsovion Interdisciplinary Research Center, National Technical University of Athens, 44200 Attica, Athens, Greece; vaioskotsios@gmail.com; 5Department of Business Administration, University of Patras, 26504 Patras, Peloponnesus, Greece; 6Department of Automation Engineering, Alexander Technological Educational Institute of Thessaloniki, 57400 Thessaloniki, Athens, Greece; tomzidr@gmail.com

**Keywords:** air pollution, coronavirus disease 2019, fine particulate matter, humidity, Italy, temperature, wind speed

## Abstract

Background: The coronavirus disease in 2019 (COVID-19) heavily hit Italy, one of Europe’s most polluted countries. The extent to which PM pollution contributed to COVID-19 diffusion is needing further clarification. We aimed to investigate the particular matter (PM) pollution and its correlation with COVID-19 incidence across four Italian cities: Milan, Rome, Naples, and Salerno, during the pre-lockdown and lockdown periods. Methods: We performed a comparative analysis followed by correlation and regression analyses of the daily average PM_10_, PM_2.5_ concentrations, and COVID-19 incidence across four cities from 1 January 2020 to 8 April 2020, adjusting for several factors, taking a two-week time lag into account. Results: Milan had significantly higher average daily PM_10_ and PM_2.5_ levels than Rome, Naples, and Salerno. Rome, Naples, and Salerno maintained safe PM_10_ levels. The daily PM_2.5_ levels exceeded the legislative standards in all cities during the entire period. PM_2.5_ pollution was related to COVID-19 incidence. The PM_2.5_ levels and sampling rate were strong predictors of COVID-19 incidence during the pre-lockdown period. The PM_2.5_ levels, population’s age, and density strongly predicted COVID-19 incidence during lockdown. Conclusions: Italy serves as a noteworthy paradigm illustrating that PM_2.5_ pollution impacts COVID-19 spread. Even in lockdown, PM_2.5_ levels negatively impacted COVID-19 incidence.

## 1. Introduction

The coronavirus 2019 disease (COVID-19) pandemic has been designated as a public emergency leading to a global health crisis [1]. While being important for COVID-19 spread mitigation, social distancing measures have significant consequences for individuals and communities [1]. The lockdowns slowed business activities, restricted transportation, and have resulted in an economic downturn affecting practically all nations [1]. However, at the same time, the traffic, transport, and industrial output restrictions and prohibitions have led to a decline in air pollution in affected countries [1,2]. 

While lockdowns have caused a decline in air pollution, this was not enough to avoid severe exceedances of air pollution in many places [1]. Italy, one of the worst-hit countries in the global coronavirus pandemic, is also one of Europe’s most polluted countries, suffering high ambient particular matter (PM) air pollution [3,4,5,6,7,8,9]. Poor air quality constitutes a well-known cause of health vulnerability and chronic inflammation, leading to a hyperactivation of the innate immune system in laboratory and human models [10,11,12,13,14,15,16,17,18]. PM pollution has been linked to an elevated risk of acute or chronic respiratory disease, respiratory infections [15], susceptibility to exacerbations, and a chronic inflammatory stimulus, even in young and healthy individuals [15,16,17,18]. Moreover, high particle pollution exposure has been associated with more medical visits and excess hospitalizations [16,17,18]. 

There is evidence supporting that PM_2.5_ and other smaller particles that remain airborne for a long time and travel long distances could act as carriers of viable virus particles, creating a suitable environment for spreading them beyond a two meters distance [19]. PMs have been incriminated in the spread of avian flu virus [19], measles [20,21], and SARS coronavirus [20]. Similarly, air pollution has been considered as one of the accelerators of COVID-19 transmission. The analysis of the microbiome in PM_10_ and PM_2.5_ in a sample of air collected in Milan (Italy) showed that the source of PMs could shape the bacterial community and influence the presence of specific bacterial groups [22,23]. In addition, Wu et al. found that an increase of only 1 μg/m^3^ in long-term average PM_2.5_ levels is associated with a significant increase of 15% in the COVID-19 death rate [24].

Moreover, pollution may influence the course of the COVID-19 pandemic by impacted comorbid conditions, such as cardiovascular diseases and diabetes, that are exacerbated by air pollution. All of these conditions are associated with poor prognosis in COVID-19 patients [24]. 

The hypothesis that poor air quality can act both as a transport vector for novel coronavirus and as a worsening factor for the health impact of COVID-19 disease, has been raised recently. Several recent studies have analyzed whether the different areas of the world with a high and rapid increase in COVID-19’s contagion were correlated to a greater level of air pollution. Italy is a great paradigm of a high-polluted country worst hit by the COVID-19 pandemic [2,25].

Studies are suggesting a relationship between PM pollution and COVID-19 spread in Italy [2]. Emerging data supported that the Italian Northern Regions, which have been the most affected by COVID-19, are also those with a high PM concentration above the legislative standards [2,25]. The extent to which PM pollution contributed to COVID-19 diffusion is needing further clarification. We aimed to assess the impact of PM_10_ and PM_2.5_ pollution on daily COVID-19 incidence across four high-polluted urbanized Italian cities: Milan, Rome, Naples, and Salerno, during the pre-lockdown and lockdown periods, after controlling for major demographic, environmental, and socioeconomic factors.

## 2. Materials and Methods

### 2.1. Data Collection

We performed a comparative analysis followed by correlation and regression analyses of the daily average concentrations of PMs with an aerodynamic diameter of 10 μm (PM_10_) or 2.5 μm (PM_2.5_) on the daily numbers of newly diagnosed COVID-19 cases across four Italian cities from 1 January 2020 to 8 April 2020, after adjustment for potential confounding demographic, environmental, and socioeconomic factors. The analyses were carried out considering a two-week earlier infection time point (lag effect). Furthermore, the PM effects were assessed separately for the pre-lockdown and lockdown periods.

The air pollution and COVID-19 case data were collected by the World’s Air Pollution website, mapping air quality globally in real time [26] and the Johns Hopkins Coronavirus Resource Center [27], respectively. There is evidence that the Italian Northern Regions and the urbanized cities of Rome and Naples, which have been the most affected by COVID-19, are also those with a high amount of atmospheric PM going above the safety standards [2]. In that context, the Italian cities of Milan, Rome, Naples, and Salerno were included in the study despite potential diverse topographical, meteorological characteristics, and pollution statutes [2].

The four cities were compared for various demographic and socioeconomic parameters, including the city population and population density, the population’s mean age, age and gender distribution, the percentage of foreigners, the number of visitors, the quality of life and health care indices, and the daily number of samples tested [28,29]. The daily average temperature, relative humidity, and wind speed data were recorded for the same period by the AccuWeather and Weather Underground websites, providing local and long-range weather conditions [30,31]. The daily average values for temperature, humidity, and wind speed were found by summing the 24-hourly measurements and dividing by the number of measurements [31,32]. The pollution conditions in Italy were quantified using the Comprehensive Air Quality index [25,26,33]. The higher safe limits for particulates in the air have been defined as a daily average of 50 μg/m^3^ for PM_10_ and a daily average of 25 µg/m^3^ for PM_2.5_, according to the World Health Organization (WHO) air quality guidelines [33].

### 2.2. Statistical Analyses

To identify differences between two independent groups, an unpaired t-test was used. Parametric data comparing three or more groups were analyzed with one-way ANOVA and Tukey’s multiple comparisons test. Non-parametric data were analyzed with the Kruskal–Wallis test and Dunn’s multiple comparison test. Spearman’s correlation was used to evaluate the monotonic relationship between two continuous variables. Stepwise multiple regression analyses was used to estimate how the demographic, meteorological, and socioeconomic variables could affect COVID-19 incidence, and how meteorological parameters could affect PM levels. The result was considered statistically significant when the *p*-value was <0.05. Data were analyzed and visualized using SPSS Statistics v.23 and Tableau Software LLC, Seattle, WA, USA, respectively.

## 3. Results

### 3.1. Demographic and Socioeconomic Parameters of Four Italian Cities

The four cities were compared for demographic and socioeconomic parameters, including each city’s population and population density, mean population age, age and gender distribution, the percentage of foreigners, the number of visitors, the quality of life and health care indices, as presented in Table 1. No significant difference was found in those aspects between Italian cities. 

### 3.2. PM_10_ and PM_2.5_ Pollution in Four Italian Cities during the Entire Study Period

Milan had a significantly higher average daily concentrations of PM_10_ and PM_2.5_ than Rome, Naples, and Salerno (Table 2). In Milan, the daily average concentration of PM_10_ exceeded the safe limits in two-fifths of the study period. Rome, Naples, and Salerno maintained safe levels of PM_10_ during the pandemic (Figure 1).

No significant difference was detected for PM_2.5_ concentrations between Rome, Naples, and Salerno. However, the average concentration of PM_2.5_ was above safe limits for all Italian cities, two to four times higher than the 24-h threshold of 25 μg/m^3^ [33]. More specifically, the daily average PM_2.5_ concentration exceeded the safe air quality standards in Milan and Naples during the entire study period (Figure 1 and Figure 2). In Rome and Salerno, the daily average PM_2.5_ concentrations exceeded the safe limits in ~88% and 98% of the study period, respectively (Figure 1 and Figure 2).

### 3.3. PM_10_ and PM_2.5_ Pollution in Four Italian Cities during the Universal Lockdown

The first lockdowns began on 21 February 2020 [34], concerning eleven municipalities of Lodi province in Lombardy. After the lockdown, there was a significant decline in average PM_10_ and PM_2.5_ concentrations in Milan (Table 2). However, the concentration of PM_2.5_ remained three times higher than the normal limits during the lockdown. On 8 March 2020, the Italian Prime Minister expanded the quarantine to all of Lombardy and 14 other northern provinces, and on the following day to all of Italy [35]. Furthermore, on the 21 March, the Italian Government closed all non-essential businesses and industries, with additional restrictions on citizens’ movement [36]. 

The universal lockdown in Italy after 21 March 21 led to a significant decline of PM_2.5_ levels in Rome, although remaining in the above-normal range. Conversely, there was no significant reduction of PM_2.5_ average levels during the lockdown in Naples and Salerno compared to the pre-lockdown period. Although Rome, Naples and Salerno maintained safe PM_10_ levels during the pre-lockdown period, there was a further reduction in PM_10_ levels during lockdown (Table 2). Figure 1 shows how levels of PM_2.5_ fell in Italy after the government imposed gradual restrictions (Figure 1).

**Figure 1 ijerph-18-05088-f001:**
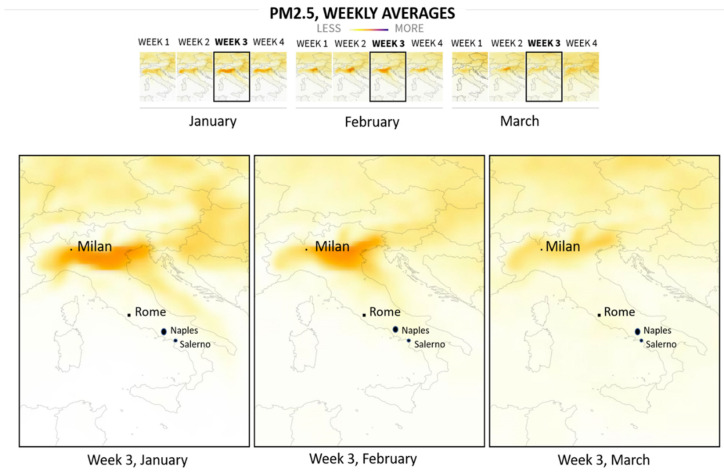
Levels of PM_2.5_ fell in four Italian cities after the government imposed gradual restrictions. 21 February 2020: the first lockdown was implemented covering eleven municipalities of the province of Lodi in Lombardy. The introduction of a nationwide lockdown was on March 9. On 21 March the paucity of industrial facilities took place. Embedded by Reuters Graphics, (2020) [37]. Copyright permission received.

Figure 2 presents the average daily concentrations of PM_10_ and PM_2.5_ applied to the daily average of new confirmed COVID-19 cases in four Italian cities for the entire study period.

**Figure 2 ijerph-18-05088-f002:**
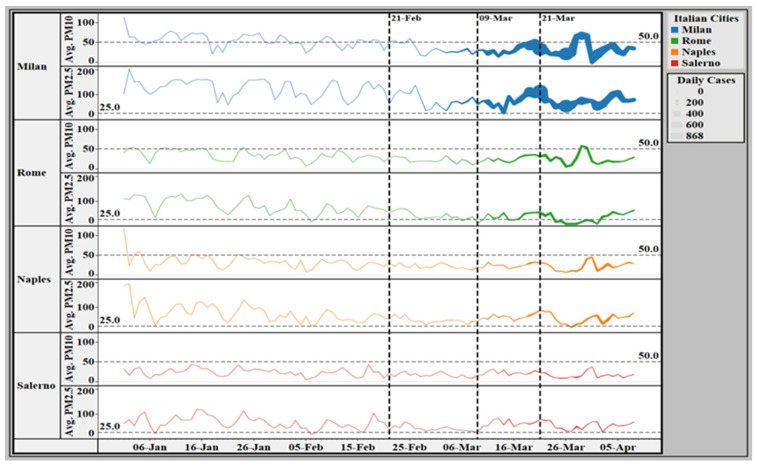
Average daily concentrations of PM_10_ and PM_2.5_ applied to the daily average of new COVID-19 cases (line width) in Milan, Rome, Naples, and Salerno for the period from 1 January 2020 to 8 April 2020.

The horizontal dashed lines represent the safe limits for PM_10_ or PM_2.5_ concentrations according to the World Health Organization air quality guidelines [32]. The vertical dashed lines represent the lockdown dates in Italy. The first line on 21 February 2020 represents the first lockdown covering eleven municipalities of the province of Lodi in Lombardy [33]. The second line on 9 March represents the lockdown to all of Italy [34]. The third line on 21 March represents the paucity of industrial facilities [35]. The line width represents the daily average of new COVID-19 cases, using the legend in the top right corner of the map. Abbreviations: Avg, average; COVID-19, coronavirus disease 2019; PM, particular matter.

### 3.4. Meteorological Conditions across Four Italian Cities

In Rome, Naples, and Salerno, the average wind speed and temperature levels were significantly lower and higher, respectively, than in Milan (Table 3). No significant difference in relative humidity was detected among the cities. However, the daily average percentage of humidity was above 60% in all four Italian cities.

A combined analysis of PM air pollution and meteorological conditions applied to the average number of confirmed new cases of COVID-19 by city is presented in Figure 3. Α significantly elevated average of new COVID-19 cases along with significantly higher PM concentrations existed in Milan as compared to the other Italian cities.

The average new confirmed COVID-19 cases by affected country is visualized as circles. Each circle was assigned a color representing each Italian city (Milan, Rome, Naples, and Salerno). The size of the circle represents the average number of new cases, using the legend in the top right corner of the map. Α. This panel represents the relationship between relative humidity and average new confirmed COVID-19 cases in each Italian city. B. This panel represents the relationship between temperature and average new confirmed COVID-19 cases in each Italian city. C. This panel represents the relationship between wind speed and average new confirmed COVID-19 cases in each Italian city. Abbreviations: Avg, average; PM, particular matter.

A multiple regression analysis was performed to predict PM_10_ or PM_2.5_ levels based on average humidity, temperature, and wind speed. A significant regression equation was found [F (3176) = 54.931, *p* < 0.001] with an *R*^2^ of 0.5. The daily average humidity, temperature, and wind speed levels were significant independent predictors of PM_10_ levels (Table 4). Similarly, the daily average humidity, temperature, and wind speed levels were significant independent predictors of PM_2.5_ levels. A significant regression equation was found [F (3176) = 62.201, *p* < 0.001] with an *R*^2^ of 0.515 (Table 4).

Together, a positive relationship between the humidity or wind speed values and daily average PM_10_ or PM_2.5_ concentrations and a negative relationship between the temperature and PM_10_ or PM_2.5_ levels was found.

### 3.5. Daily COVID-19 Incidence across Four Italian Cities

The cumulative and average daily COVID-19 incidence across four Italian cities is presented in Table 5. Milan had the highest number of newly confirmed cases in a single day, followed by Rome, Naples, and Salerno. The daily COVID-19 incidence was significantly higher during the lockdown period than in the pre-lockdown period in all Italian cities. No significant correlation between the city population and the daily sampling rate for COVID-19 was detected (r = 0.112, *p* = 0.133).

### 3.6. Investigating the Effect of PM Pollution on COVID-19 Incidence after Adjusting for Demographic, Environmental and Socioeconomic Factors

Given that the median incubation period for COVID-19 has been estimated at approximately five days (95% CI, 4.5 to 5.8 days) and the vast majority (97.5%) of those who develop symptoms will do so within 11 days (95% CI, 8.2 to 15.6 days) of infection [38], a lag of 14 days was used for the analyses.

The average concentrations of PM_10_ were below the safe limits for all the Italian cities, except Milan; consequently, the effect of PM_10_ pollution on COVID-19 incidence could not be effectively evaluated.

The daily COVID-19 incidence was positively correlated with the daily PM_2.5_ (r = 0.6, *p* < 0.001) concentrations. To investigate whether PM_2.5_ exposure was associated with COVID-19 incidence, a multiple stepwise regression model with appropriate adjustment for environmental, demographic, and socioeconomic factors of four Italian cities was performed. The effects of several factors including the daily average temperature, humidity, wind speed levels, the city population, population density, gender and age distribution, percentage of foreigners, number of visitors, quality of life, health care indices, and the daily sampling rate in each city (independent variables) on daily COVID-19 incidence (dependent variable) were assessed separately for the pre-lockdown and lockdown periods. 

Based on the factor analysis scores during the pre-lockdown period, the predictor variables of the daily PM_2.5_ levels and the daily sampling rate explained 88.6% of the total variance in the regression model (Table 6, Model 1).

A multiple stepwise regression analysis was performed for daily newly diagnosed COVID-19 cases during universal quarantine (21 March 2020–8 April 2020) using the aforementioned independent variables (Table 6, Model 2). The predictor variables of daily PM_2.5_ levels, the mean population’s age, and the population density explained 89.6% of the total variance in this regression model, while all the other variables were excluded from the analysis. 

## 4. Discussion

Italy is one of the most highly polluted countries in Europe. Furthermore, Italy registered more deaths in 2020 than in any other year since World War II, attributed to COVID-19. Milan had significantly higher average daily PM_10_ and PM_2.5_ levels than Rome, Naples, and Salerno. Rome, Naples, and Salerno maintained safe levels of PM_10_ during the pandemic. PM_10_ levels fell further during the lockdown in all Italian cities. However, in Milan, the average daily concentration of PM_10_ exceeded the safe limits in two-fifths of the study period. We found that PM_2.5_ pollution dropped significantly during lockdowns in Milan and Rome, but not in Naples and Salerno. Nevertheless, severe air pollution events were not avoided by reduced anthropogenic activities during the COVID-19 outbreak. PM_2.5_ concentrations remained much higher than the safe limits in all Italian cities. Our findings broadly support the work of recent studies linking PM pollution with COVID-19 incidence. More specifically, the current found that the PM_2.5_ levels and sampling rate were strong predictors of COVID-19 incidence during the pre-lockdown period. The PM_2.5_ levels, population’s age and density strongly predicted COVID-19 incidence during the lockdown. 

PM_2.5_ pollution can be either human-made or naturally occurring. Traffic is the strongest primary source for PM_2.5_ in Italy, together with secondary inorganic and organic aerosol formation during spring and summer [39]. The increased industrialization in Italian cities has been strongly related to PM_2.5_ pollution [39]. Among many other sectors, the industry sectors and transport were the most hard-hit due to lockdown. Road and air transport came to a halt as people were not allowed or were hesitant to travel. However, it seems that although lockdowns caused the decline of air pollution, this was not enough to avoid severe PM_2.5_ pollution events in Italy.

Meteorological parameters could impact PM variations. The current study found a positive relationship between the humidity or wind speed with daily average PM concentrations and a negative relationship between the temperature and PM levels. It has been previously reported that the highest PM_2.5_ values were observed under a relative humidity of 70%–90% and low wind speed conditions [40,41]. An increase in relative humidity could aggravate PM pollution through physical and chemical processes, affecting the gas-to-particle conversion rate and wet or dry deposition [42,43,44,45,46]. The average daily percentage of humidity was found to be high in all four Italian cities. In particular, it was higher than 60%, a threshold that is considered uncomfortable for many people. Moreover, it is accepted that PMs descend when the wind velocity increases [14,40,47,48,49,50]. Minimum PM values occur when the wind velocity reaches 1.0–2.0 m/s [14]. A much-debated question is whether temperature correlates with PM pollution [7,50,51]. We found a negative correlation between the temperature and PM_2.5_ or PM_10_ levels. More specifically, we found that PM_10_ and PM_2.5_ levels decreased 2.8 and 6 μg/m^3^ for each °C of temperature. Supportive evidence shows that PM emissions increased exponentially as temperature decreased, suggesting a negative correlation between temperature and PMs [7]. A possible explanation for this finding is that when the temperature is higher, the air convection at the lower surface is stronger, which benefits the upward transport of particular matter [52]. However, other reports showed a positive correlation, or no correlation at all [5,50,51].

Italy has a Mediterranean climate with cold, wet, and windy winters, especially in the North. Milan constitutes one of the biggest air pollution hotspots in Europe due to its unfavorable meteorological conditions and its topography, causing the air trapping of pollutants [2,53]. Our data showed that Milan potentially had favorable meteorological conditions for PM generation amid the study period, having significantly higher wind speed and lower temperature conditions than the rest of the cities in Central and Southern Italy. 

There has been a lively debate over whether climatic conditions modulate virus infectivity and spreading [54]. This study suggested that although meteorological conditions impact PM pollution, they could not predict the epidemic situation.

At the same time, a significantly elevated average of daily newly diagnosed COVID-19 cases existed in high-polluted Milan compared to other Italian cities. The number of new confirmed COVID-19 cases was increased for all cities during the lockdown period. Our findings broadly support the work of a few recent studies linking PM pollution with COVID-19 spreading both in pre-lockdown and lockdown periods in Italy [55,56]. We found that higher PM_2.5_ concentrations were linked to a higher daily COVID-19 incidence across Italy. Moreover, the analyses showed that the PM_2.5_ pollution and the daily sampling rate for COVID-19 could independently predict daily COVID-19 incidence during the pre-lockdown period. Furthermore, the average daily PM_2.5_ levels and the mean age and density of the city population strongly predicted the daily COVID-19 incidence during the universal lockdown. However, a lockdown-related reduction in PM_2.5_ levels was not accompanied by a reduction in virus transmission. The question then arises whether and how COVID-19 as an acute disease is associated with immediate air pollution. A possible explanation is that long-term high PM_2.5_ levels might add to high COVID-19 incidence; thus, our findings represent a “snapshot” of chronic exposure to air pollution.

PM pollution should be considered in a long-term, chronic perspective [25,55,56,57], as PMs travel deeply via airways reaching the lung parenchyma and may remain in the deep parts of the lung for years. This extended PM exposure could induce modifications to immunity [13,25]. Short-term changes in the air quality may not be sufficient to break this effect [25]. Fattorini et al. reported that PM_2.5_ correlated with cases of COVID-19 in up to 71 Italian provinces, providing evidence that chronic exposure to atmospheric contamination may represent a favorable context for the spread of the virus [25]. Therefore, prolonged PM_2.5_ exposure could be one of the main reasons for Italy’s alarming spreading that was observed despite reductions in atmospheric PM pollution during lockdown [25].

PM’s inhalation led to pulmonary inflammation characterized by alveolar macrophage and neutrophil proliferation and activation [5,6,7,8,9,14,55,56,57]. The activation of monocytes determines the reactive oxygen species production and cytokine generation that play a critical role in fibrosis development. Although the fibrotic process is essential for tissue repair, it can compromise the lung structure and pulmonary physiological functions, when it exceeds a certain threshold [14,16,57]. A crucial pattern of PM_2.5_-induced lymphatic endothelial cell cytotoxicity in the human respiratory system has also been reported [57].

The geographic differences in the number of COVID-19 cases likely reflect the interaction between several epidemiologic and population-level factors, including the timing of COVID-19 introductions, population density, age distribution, diagnostic testing capacity, and public health reporting practices [58]. A key strength of the present study was that it comprehensively examined the impact of several demographic, meteorological, and socioeconomic variables on COVID-19 incidence. We documented that apart from the PM_2.5_ pollution, the sampling rate during the pre-lockdown period, and the population’s density and age during the lockdown, were predictors of COVID-19 incidence.

Data from several studies suggest that the daily testing of the population provides added benefit in various infectiousness contexts [54,59]. Furthermore, the importance of population density has been ascertained in modeling the spread of diseases and this parameter should be explicitly included in transmission models that predict the impacts of COVID-19 [59]. Our results corroborate the findings of a great deal of the previous work regarding the necessity of age-based measures that isolate older people. Severity estimates from COVID-19 suggested that mainly those aged ≥70 years were disproportionately affected by the pandemic, a finding consistent among regions [59]. A context that has recently changed after elderly people have been vaccinated for COVID-19.

The COVID-19 pandemic constitutes a multifactorial problem requiring multifactorial responses. Our study provides some novel insights into COVID-19 incidence after adjusting for several relevant confounding factors concerning four Italian cities. However, it has some limitations that need to be acknowledged. Firstly, our analysis did not include all Italian cities. Secondly, this study did not examine the effect of other air pollutants such as carbon monoxide, nitrogen dioxide, and sulfur dioxide, which may increase the risk of respiratory tract infections. Our findings should be put into context with other critical factors such as increased pathogen virulence, the Government’s prevention policies, and the wide differences among regional preventive care. We recently highlighted the profoundly negative impact of bad politics on crisis management as a bad policy could be more dangerous than a virus [54]. Continued epidemiological and experimental studies are needed to evaluate the role of atmospheric pollution in specific populations and provide more critical information for better preparedness policies in cases of pandemics.

## 5. Conclusions

Italy is one of the worst hit European countries by the COVID-19 pandemic, and a noteworthy example illustrating PM_2.5_ pollution significantly impacts COVID-19 incidence. While lockdowns have caused a decline in air pollution, this was not enough to avoid severe exceedances of air pollution in all examined Italian cities. Our findings indicate that Milan had mainly favorable meteorological conditions for PM generation and stagnation amid the pandemic. However, although meteorological conditions exert an impact on PM pollution, they could not predict the epidemic situation. On the other hand, we found that daily PM_2.5_ levels predicted COVID-19 incidence. The PM_2.5_ levels and sampling rate were strong predictors of COVID-19 incidence during the pre-lockdown period. The PM_2.5_ levels, population age, and density strongly predicted COVID-19 incidence during the lockdown. These findings represent a “snapshot” of the health impact of chronic exposure to air pollution. Continued studies are needed to evaluate the role of air pollution in specific populations and offer specific recommendations for better policies. In that context, global communities must unite for pollution control action plans if we are to prevent any further global–scale pandemics, given that the majority of infections are linked to the collapse of the environment. 

## Figures and Tables

**Figure 3 ijerph-18-05088-f003:**
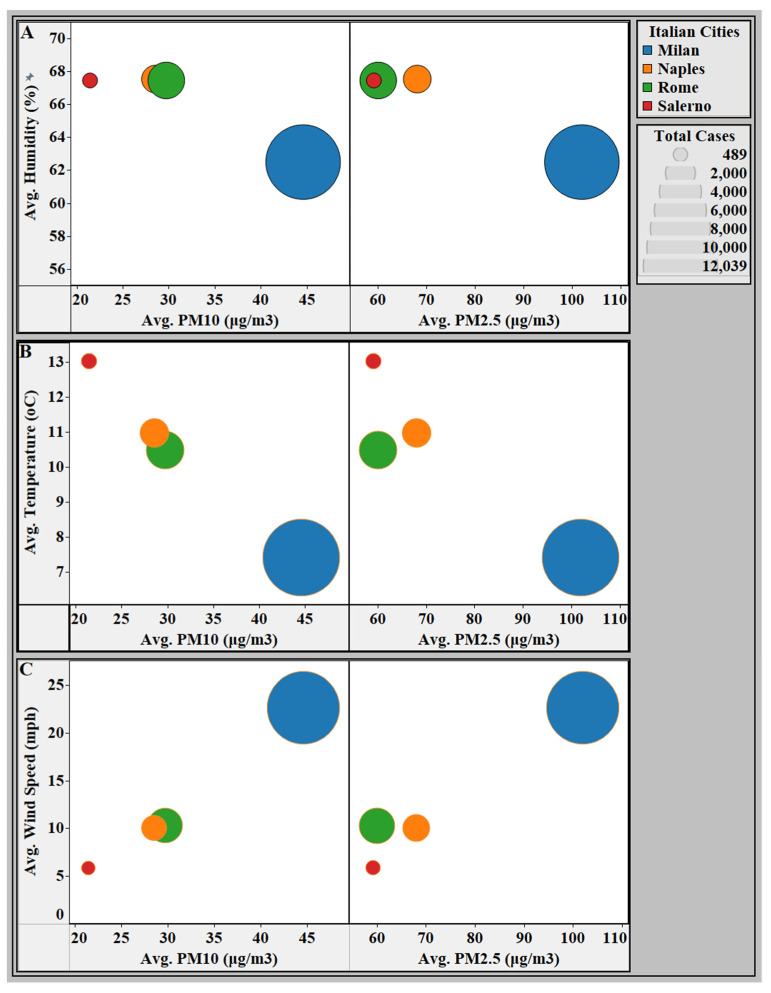
Comparison analysis between the average PM_10_ or PM_2.5_ concentrations and climatological conditions applied to the average number of confirmed new cases of COVID-19 by city (January 2020–April 2020).

**Table 1 ijerph-18-05088-t001:** Comparisons of demographic and socioeconomic parameters between four high-polluted Italian cities (Milan, Rome, Naples, and Salerno).

Various Investigated Aspects	Milan	Rome	Naples	Salerno	*p*-Value
Population (N)	3,250,315	4,342,212	3,084,890	1,098,513	NS
Population density (people/Km^2^)	206.3	809.6	2616.8	221.7	NS
Males (%)	48.5	47.9	48.5	48.9	NS
Females (%)	51.5	52.1	51.5	51.1	NS
Foreigners (%)	14.5	12.8	4.4	5.2	NS
Mean population age (years)	44.8	44.5	41.3	43.6	NS
Population age structure					
0–17 years (% of population)	522,975 (16.09)	700,833 (16.14)	571,938 (18.54)	178,728 (16.27)	NS
18–64 years (% of population)	1,991,469 (61.27)	2,702,159 (62.23)	1,951,192 (63.25)	689,536 (62.77)	NS
65+ years (% of population)	735,871 (22.64)	939,220 (21.63)	561,760 (18.21)	230,249 (20.96)	NS
Visitors, 2019 (N)	9,291,198	7,046,098	7,247,964	2,098,781	NS
Quality of Life Index	117.43	110.75	102.39	145.73	NS
Health Care Index	71.57	81.48	56.01	54.17	NS

All parameters were tested with the Kruskal–Wallis test as the sum of cases weight was less than five. Abbreviations: NS, non-significant.

**Table 2 ijerph-18-05088-t002:** Average PM_10_ and PM_2.5_ concentrations in Milan, Rome, Naples, and Salerno during the pre-lockdown and lockdown study periods (1 January 2020–8 April 2020).

	Italian Cities	Milan	Rome	Naples	Salerno
PM_10_ (μg/m^3^) Safe level: <50)	Entire study period	45.1 ± 18.5 *^,^**^,^***	29.7 ± 12.7 *^,#^	28.6 ± 15.0 **^,^^	21.4 ± 9.0 ***^,#,^^
	Pre-lockdown period	47.3 ± 18.2	31.0 ± 12.3	30.1 ± 14.9	22.6.7 ± 8.8
	Lockdown period	34.6 ± 16.8	24.4 ± 13.6	21.6 ± 11.2	16.1 ± 7.7
	*p*-value ^1^	0.010	0.050	0.025	0.005
PM_2.5_ (μg/m^3^) (Safe level: <25)	Entire study period	102.0 ± 38.3 *^,^**^,^***	60.1 ± 32.3 *	68.1 ± 30.8 **	59.2 ± 21.7 ***
	Pre-lockdown period	108.4 ± 39.1	66.2 ± 31.4	70.9 ± 32.2	60.0 ± 22.8
	Lockdown period	73.3 ± 15.1	32.4 ± 19.0	55.4 ± 18.9	52.9 ± 14.6
	*p*-value ^1^	<0.001	<0.001	0.050	0.046

Data are expressed as mean ± SD. One-Way ANOVA and Tukey’s multiple comparisons test were used to determine pairwise differences of means of PM_10_ or PM_2.5_ between Italian cities for each period: *,**,***, # *p* < 0.001, ^ *p* = 0.002. ^1^ Comparison between pre- and lockdown daily average PM levels in each Italian city.

**Table 3 ijerph-18-05088-t003:** Comparison of meteorological conditions among Milan, Rome, Naples, and Salerno (1 January 2020–8 April 2020).

Italian Cities	Milan	Rome	Naples	Salerno	*p*-Value
Daily average Humidity (%)	62.5 ± 22.4	67.4 ± 13.0	67.5 ± 14.2	67.4 ± 14.7	NS
Daily average temperature (°C)	7.4 ± 3.5 *^,^**^,^***	10.5 ± 2.8 *^,#^	11.0 ± 2.5 **^,^^	12.0 ± 2.2 ***^,#,^^	*p* < 0.001
Daily average wind speed (mph)	22.6 ± 31.3 *^,^**^,^***	10.3 ± 4.7 *	10.0 ± 5.0 **	12.7 ± 10 ***	*p* < 0.001

Data are expressed as mean ± SD. One-Way ANOVA and Tukey’s multiple comparisons test: *,**,***, ^#^
*p* < 0.001, ^^^
*p* = 0.044. Abbreviations: NS, non-significant.

**Table 4 ijerph-18-05088-t004:** Multiple linear stepwise regression model for the prediction of PM_10_ (Model 1) and PM_2.5_ levels (Model 2) based on meteorological parameters (January 2020–8 April 2020).

**Model 1**	**Unstandardized Coefficients**		**Standardized Coefficients**	**t**	**Sig.**	**Collinearity Statistics**	
	**B**	**Std. Error**	**Beta**			**Tolerance**	**VIF**
(Constant)	34.246	6.603		5.187	<0.001		
Daily average Humidity (%)	0.312	0.072	0.323	4.359	<0.001	0.535	1.870
Daily average temperature (°C)	−2.835	−2.80	−0.571	−10.138	<0.001	0.925	1.081
Daily average wind speed (mph)	0.268	0.057	0.360	4.744	<0.001	0.509	1.964
Dependent Variable: PM_10_ levels The PM pollution of Milan, Rome, Naples, and Salerno included in the analysis. R = 69.5%, R^2^ = 50.0%, R^2^ (Adjusted) = 47.5%							
**Model 2**	**Unstandardized Coefficients**		**Standardized Coefficients**	**t**	**Sig.**	**Collinearity Statistics**	
	**B**	**Std. Error**	**Beta**			**Tolerance**	**VIF**
(Constant)	55.828	14.695		3.799	<0.001		
Daily average Humidity (%)	0.921	0.159	0.415	5.780	<0.001	0.535	1.870
Daily average temperature (°C)	−6.008	0.623	−0.527	−9.651	<0.001	0.925	1.081
Daily average wind speed (mph)	0.829	0.126	0.485	6.588	<0.001	0.509	1.964
Dependent Variable: PM_2.5_ levels The PM pollution of Milan, Rome, Naples, and Salerno included in the analysis. R = 71.7%, R^2^ = 51.5%, R^2^(Adjusted) = 50.6%							

**Table 5 ijerph-18-05088-t005:** Total COVID-19 cases and the daily average of new COVID-19 cases in Milan, Rome, Naples, and Salerno (January 2020–08 April 2020).

Italian Cities	Milan	Rome	Naples	Salerno
Total cases up to 8 April 2020	12039 *^,^**^,^***	2910 *	1668 *	489 ***
Daily average new COVID-19 cases				
Entire study period	268 ± 226 *^,^**^,^***	66 ± 53 *^,#,^^	37 ± 37 **^,#,&^	11 ± 11 ***^,^,&^
Pre-lockdown period	173 ± 170	34 ± 34	17 ± 16	5 ± 5
Lockdown period	409 ± 156	112 ± 29	67 ± 34	20 ± 15
*p*-value ^3^	<0.001	<0.001	<0.001	<0.001

Data are expressed as mean ± SD. One-Way ANOVA and Tukey’s multiple comparisons test: *,**,***, ^,^&^
*p* < 0.001, ^#^
*p* = 0.044. ^3^ Comparison between pre- and lockdown daily average PM levels in each Italian city. Abbreviations: COVID-19, coronavirus disease 2019.

**Table 6 ijerph-18-05088-t006:** Multiple regression analysis for daily COVID-19 incidence during the pre-lockdown period (24 February 2020–21 March 2020), considering a two-week lag in COVID-19 test results.

**Model 1**	**Unstandardized Coefficients**		**Standardized Coefficients**	**t**	**Sig.**	**Collinearity Statistics**	
	**B**	**Std. Error**	**Beta**			**Tolerance**	**VIF**
(Constant)	32.319	8.728		3.703	<0.001		
Daily number of samples tested	0.26	0.001	0.988	28.876	<0.001	0.805	1.242
Daily average PM_2.5_ (μg/m^3^)	0.383	0.116	0.122	3.305	0.001	0.805	1.242
Dependent Variable: Daily average of new COVID-19 cases during the pre-lockdown period. The Italian cities of Milan, Rome, Naples, and Salerno included in the analysis. Excluded variables: PM_10_ (μg/m^3^), Daily average humidity (%), Daily average temperature (°C), Daily average wind speed (mph), Population (n), Population density (people/km^2^), Males (%), Females (%), Foreigners (%), Visitors (n), Mean age of population (years), Quality of life index, Health care index. R = 94.1%, R^2^ = 88.6%, R^2^ (Adjusted) = 83.3%.							
**Model 2**	**Unstandardized Coefficients**		**Standardized Coefficients**	**t**	**Sig.**	**Collinearity Statistics**	
	**B**	**Std. Error**	**Beta**			**Tolerance**	**VIF**
(Constant)	−4.070	358.747		−12.098	<0.001		
Daily average PM_2.5_ (μg/m^3^)	1.750	0.322	0.338	5.442	<0.001	0.689	1.452
Mean age of population (years)	90.747	8.195	0.726	11.073	<0.001	0.617	1.621
Population density (people/km^2^)	0.107	0.013	0.598	8.569	<0.001	0.546	1.832
Dependent Variable: Daily average of new COVID-19 cases during the lockdown period. The Italian cities of Milan, Rome, Naples, and Salerno were included in the analysis. Excluded variables: PM_10_ (μg/m^3^), Daily average humidity (%), Daily average temperature (°C), Daily average wind speed (mph), Population (N), Males (%), Females (%), Foreigners (%), Visitors (n), Quality of life index, Health care index. Sampling (n), R = 90.5%, R^2^ = 82.0%, R^2^ (Adjusted) = 81.2%.							

## Data Availability

The data that support the findings of this study are available on request from the corresponding author, O.S.K.

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
