# Peer review of "PM2.5 Pollution Strongly Predicted COVID-19 Incidence in Four High-Polluted Urbanized Italian Cities during the Pre-Lockdown and Lockdown Periods"

_ijerph, 2021, doi:10.3390/ijerph18105088_

Round 1

Reviewer 1 Report

This paper is lack of writing norms, for example, the background, methods and conclusions are listed in the abstract, separately; the introduction only uses a few short paragraphs to explain; the discussion part describes the listing and statement of the data in each table, lack of in-depth analysis and causes of data changes; the font size of the body part is different, the spacing between the lines is different; the conclusion part should be extended.

Author Response

COMMENTS FROM REVIEWER 1:

  1. This paper is lack of writing norms, for example, the background, methods and conclusions are listed in the abstract, separately;

RESPONSE: We are very grateful for the effort you dedicated to reviewing our submission, as well as your favorable comments. We paid heed to your advice and suggestions, and the manuscript has been massively revised. We hope the revised version of our manuscript will satisfy your concerns.

In the revision, the abstract has been restructured as a single paragraph according to the writing norms (page 1, lines 24-44).

  1. the introduction only uses a few short paragraphs to explain;

RESPONSE: Thank you for this remark. Per your suggestion, we have added four paragraphs in this section (Page 2, lines 55-73), supported by references 19-24.

  1. the discussion part describes the listing and statement of the data in each table, lack of in-depth analysis and causes of data changes;

RESPONSE: Thank you for this valuable direction. We have revised the discussion section as advised, and have undertaken an in-depth analysis of the causes of data changes.

  1. the font size of the body part is different,

RESPONSE: Thank you for the comment. In the revision, we prepared our submission according to the specifications set out in the Author Guidelines.

  1. the spacing between the lines is different;

RESPONSE: Thank you for this point. We have reformed the manuscript according to the writing norms, as suggested.

  1. the conclusion part should be extended.

RESPONSE: Thank you for the comment. Per your suggestion, the conclusion part has been extended (page 13, lines 424-433).

We appreciate all of your insightful comments. We found them quite helpful as we approached our revision. We hope you find these revisions rise to your expectations.

Reviewer 2 Report

It is well written and clear paper. In my view, the authors should include pictures, such as the attached one from the work of Karl et al. (2019) doi: 10.5194/acp-19-1721-2019. The demonstration and illustration of the impact will get improved, so the visibility of this work and the resulting referencing.

Author Response

COMMENTS FROM REVIEWER 2

  1. It is well written and clear paper.

RESPONSE: We sincerely thank you for your kind words about our paper. We are delighted to receive positive feedback from you.

  1. In my view, the authors should include pictures, such as the attached one from the work of Karl et al. (2019) doi: 10.5194/acp-19-1721-2019. The demonstration and illustration of the impact will get improved, so the visibility of this work and the resulting referencing.

RESPONSE: Thank you for this direction. Per your suggestion, in the revision, we have introduced Figures 1-3, expecting to raise the visibility of this work.

We really thank you for taking the time and energy to help us improve this paper. We very much appreciated your encouraging and insightful comments.

Reviewer 3 Report

I did not find full tables in the current pdf manuscript, the authors also need to revise these points of *p<0.001, **p<0.001, ***p<0.001, #p<0.001,. Why did you consider humidity if it is not significant and to what extent temperature cause to increase in the PM10 and PM2.5? The authors should also consider the age range of the population besides only groups. The authors also need to define their data in graphical presentations rather than tables.

Author Response

COMMENTS FROM REVIEWER 3

  1. I did not find full tables in the current pdf manuscript, the authors also need to revise these points of *p<0.001, **p<0.001, ***p<0.001, #p<0.001.

RESPONSE:  We greatly appreciate your constructive comments and suggestions that helped improve the manuscript. We have carefully studied them and tried to revised our paper accordingly. Per your suggestion, in the revision we have revised the points you mentioned.

  1. Why did you consider humidity if it is not significant,

RESPONSE: Thank you for this comment. Although no significant difference in relative humidity was detected among the four Italian cities, all the meteorological parameters, including humidity, have been reported to impact PM variations. Indeed, we found that humidity was a significant independent predictor of PM10 and PM2.5 levels (Table 4).

  1. and to what extent temperature cause to increase in the PM10 and PM2.5?

RESPONSE:  Thank you for this comment. We discuss this issue on page 11, lines 307-318.

  1. The authors should also consider the age range of the population besides only groups.

RESPONSE: Thank you for this comment. Per your recommendation, in the revision we present the age structure of the population by major age groups in Table 1. No significant difference was found in age groups between the four Italian cities.

  1. The authors also need to define their data in graphical presentations rather than tables.

RESPONSE: Thank you for this remark. In the revision, we have introduced three graphical presentations (Figures 1-3) to define our data further.

We appreciate you taking the time to offer us your insights related to the paper. We hope you find these revisions rise to your expectations.